# Strategic Considerations in Designing Food Solutions for Seniors

**DOI:** 10.3390/foods14030396

**Published:** 2025-01-25

**Authors:** Leehen Mashiah, Anais Lavoisier, Shannon Gwala, Andrea Araiza Calahorra, Carmit Shani Levi, Rune Rødbotten, Paula Varela, Anwesha Sarkar, Andre Brodkorb, Didier Dupont, Uri Lesmes

**Affiliations:** 1Laboratory of Chemistry of Foods and Bioactives, Department of Biotechnology and Food Engineering, Technion—Israel Institute of Technology, Haifa 3200003, Israel; leehen@campus.technion.ac.il (L.M.); shanilc@bfe.technion.aci.il (C.S.L.); 2INRAE—UMR STLO, 85 rue de Saint Brieuc, 35000 Rennes, France; anais.lavoisier@inrae.fr (A.L.); didier.dupont@inrae.fr (D.D.); 3Moorepark Food Research Centre, Teagasc, Moorepark, P61 E224 Fermoy, Cork, Ireland; shannon.gwala@teagasc.ie (S.G.); andre.brodkorb@teagasc.ie (A.B.); 4Food Colloids and Bioprocessing Group, School of Food Science and Nutrition, University of Leeds, Leeds LS2 9JT, UK; a.araizacalahorra@leeds.ac.uk (A.A.C.); a.sarkar@leeds.ac.uk (A.S.); 5Nofima AS., P.O. Box 210, 1431 Ås, Norway; rune.rodbotten@nofima.no (R.R.); paula.varela.tomasco@nofima.no (P.V.)

**Keywords:** elderly nutrition, nutritional deficiencies, functional foods, tailored foods, protein fortification, food product development, SWOT

## Abstract

The demographic shift towards an aged population calls for targeted nutrition strategies to support healthy aging and bridge the gap between life expectancy and a healthy life span. Older adults face various nutritional deficiencies, particularly in protein, vitamins (B12, D), minerals (calcium, iron), and dietary fiber. This work delves into the EAT4AGE project efforts that strategically aimed to develop age-oriented food products (European Joint Programming Initiative “A Healthy Diet for a Healthy Life” JPI HDHL). Currently, manufacturing of age-tailored food products presents significant complexities, from challenges of commercialization to the generation of acceptable and palatable food choices. As a first step, a literature-based comprehensive checklist has been developed to facilitate product development. This tool provides an integrated approach, ensuring that all critical aspects of product development are addressed systematically. Secondly, we describe the application of the tool in the development of a series of products, such as plant-based protein-rich cereals, reformulated dairy products, processed meat, and enriched spreads; all combining high nutritional values with adaptations to the physiological and sensory needs of seniors. Overall, this work offers insight into the current needs of seniors and a tool for product development that can be utilized for prospective product development, such as the ones detailed herein. Thus, the EAT4AGE hopes to set an example that will stimulate the fabrication of effective, well-received nutritional solutions, ultimately improving health outcomes for older adults.

## 1. Introduction

The global rise in life expectancies is accompanied by a health gap between life span and health span [1]. This could be mitigated by various interventions and food solutions that could help secure healthy aging and human well-being [2]. In fact, aging is worryingly linked with malnutrition, which can profoundly affect the well-being of older adults and increase their risk of chronic diseases and premature mortality [3,4,5,6,7]. Nutrition-specific interventions could offer an avenue of possibilities to mitigate malnutrition while considering age-related physiological changes [2,8,9,10,11]. Thus, one can note increasing research efforts to develop foods for this important demographic, as evidenced by its occurrence in articles indexed by Clarivate Web of science between 2000–2024 and retrieved during July 2024 with key terms such as “foods for elderly/seniors/older adults”, “elderly nutrition”, “rational food design”, and “healthy aging” (Figure 1). The majority of the articles are in the fields of Nutrition Dietetics, Public Health, Geriatrics/Gerontology and Food Science Technology, comprising approximately 69% of the published research. These soaring efforts focus on the third global Sustainable Development Goal (SDG) of good health and well-being [12], yet the low ratio between the number of publications and number of citations indicates that the field is still in its infancy. Thus, there is an imminent need to develop strategies for targeted food product development which is also supported by numerous funding agencies around the globe.

In order to help advance systematic projects in the field, a comprehensive approach such as a SWOT (strengths, weaknesses, opportunities, and threats) analysis can be applied to catalyze development of innovative and tailored nutritional solutions for older adults (Figure 2).

In respect to strengths, the senior nutrition market is experiencing significant growth. According to Verified Market Research, the Global Elderly Nutrition Market was valued at USD 17.62 billion in 2018 and is expected to reach USD 28.89 billion by 2026, with a compound annual growth rate (CAGR) of 6.39% from 2019 to 2026 [13]. This viable commercial target is also supported by a growing body of scientific research into nutrition of older adults (as indicated in Figure 1), including insights into the digestive fate of foods in seniors, their food preferences, and the efficacy of foods and nutraceuticals as tools to direct healthy aging [2,3,7,10,11,14,15,16]. Moreover, it is essential to consider sensory decline, including loss of taste and odor perception in older consumers, which can significantly impact their food preferences [17]. This is also supported by reports on nutritional gaps in seniors and updated suggested nutritional recommendations [5,6,7,18]. The nutritional solutions sector for older adults faces limited competition due to significant challenges, including complex nutritional needs and stigma surrounding products designed for seniors. Companies that can effectively develop innovative and acceptable products tailored to seniors may find themselves well positioned in the expanding “silver economy.” This market supports the creation of age-appropriate nutritional solutions and aligns with SDG3, which emphasizes good health and well-being for all age groups [12].

Regarding weaknesses, contrary to the growing demographics of seniors and expanding knowledge base, current efforts to rationalize food production have yet to address age-related physiological decline [2], consumer adherence to eating patterns [19] and the growing medical complexities of older adults, once they experience the health gap [1]. In terms of opportunities, this opens opportunities for endeavors that could capitalize on progress in food technologies, precision nutrition and functional foods as well as the growing consumer demand for healthier food choices. As for threats, these opportunities should be carefully balanced against the challenges of marketing, consumer adoption of innovative foods and the need to establish their clinical efficacy and overall cost-effectiveness. From a regulatory standpoint, products targeting vulnerable populations like seniors face more stringent requirements for health claims and nutritional content, necessitating thorough testing and validation. In addition, the development of age-tailored foods is threatened by nutritional guidelines across cultures and regions, which may add another layer of complexity to product development and distribution. Successfully navigating SWOT determinants (Figure 2) is key to developing products that not only meet the nutritional needs of seniors but undergo successful commercialization and consumer adoption towards impacting public health. Thus, the EAT4AGE project, conducted under the ERA-HDHL initiative, sought to follow a general framework (outlined in Figure 3) that entails four main stages of development: first, identifying consumer gaps, followed by ideation and screening of possible solutions, and then prototyping a product towards scale-up, commercialization and launch of a new food choice.

## 2. Identifying the Gaps—Nutritional Deficiencies in Seniors

Aging is well documented to be accompanied by physiological changes, including age-specific changes in the gut microbiota [8,20,21,22,23]. This issue is compounded by changes in eating patterns (e.g., reduced appetite and difficulty chewing), sociological and psychological changes (e.g., loneliness and depression) underlying the dietary choices of older adults. Sensory decline, particularly in taste and odor perception, is a well-documented phenomenon associated with aging [17]. This deterioration significantly influences dietary habits and nutritional intake among the elderly, as it reduces the appeal and enjoyment of food, potentially leading to inadequate nutrition. Elderly nutritional status can be viewed from the epidemiological perspective, i.e., timed analysis of documented surveys of nutritional inadequacies and deficiencies or poor dietary choices which may span from energy and macronutrient intake to gaps in micronutrients and bioactive non-nutrients; all crucial for maintaining health and well-being in older adults [5,24,25,26,27,28,29,30].

In respect to energy and macronutrients, seniors’ caloric intake has been documented to decline with age, and protein has been identified as a key target for supplementation deficiency [5,7,18,31,32,33]. In fact, research by Moore et al. demonstrates that older adults, due to age-related anabolic resistance, require approximately double the per-meal protein dose compared to younger individuals to achieve similar muscle protein synthesis [34]. This highlights the need for higher protein intake in the elderly to combat sarcopenia and maintain muscle mass and function, which can potentially mitigate through targeted public nutrition initiatives [35,36]. The decline in digestive enzymatic activity, such as pepsin and pancreatic enzymes, further complicates the digestion and absorption of these macronutrients.

Seniors have also been found to face deficiencies in essential vitamins and minerals [5,26,28,29,30]. Recent reviews and surveys indicate vitamin B12 and vitamin D deficiencies are prevalent, affecting approximately 15% of the elderly due to malabsorption or inadequate dietary intake [29]. Alarmingly, more than 40% of the European population has insufficient vitamin D levels (below 20 ng/mL), with over 13% facing acute deficiency (less than 12 ng/mL) [37]. These deficiencies that may be accompanied by low levels of minerals such as calcium, iron, selenium, zinc, copper, iodine and magnesium are common and contribute to conditions like anemia, compromised immune function, osteoporosis and delayed wound healing. Ensuring adequate intake of vitamin D and calcium is crucial for maintaining bone health, supporting muscle function, and potentially reducing the risk of various chronic diseases, including cardiovascular, neurodegenerative, and autoimmune disorders [37]. Vitamins A, C, and E have shown benefits for muscle strength and sarcopenia prevention [38,39].

Non-nutrient phytochemicals and antioxidants are essential for protecting against oxidative damage and age-related diseases, such as cognitive decline and age-related macular degeneration (AMD) [40,41,42]. Key compounds like carotenoids (lutein, zeaxanthin, β-carotene), flavonoids, and polyphenols are crucial for cognitive function, eye health, and cellular protection in older adults [43]. However, seniors often struggle to obtain sufficient amounts of these nutrients due to reduced fruit and vegetable consumption, decreased appetite, dental issues, and limited access to fresh produce, especially for those with mobility challenges. Poor diet quality and malnutrition further exacerbate these deficiencies, while food processing can diminish the availability of these beneficial compounds in prepared meals [6,44,45,46,47,48].

Similarly, fiber, which is considered both a nutrient and an anti-nutrient in certain foods, is often deficient in the diets of seniors. This deficiency is common due to factors such as decreased appetite, dental problems, and reduced gastrointestinal motility, which often lead to lower consumption of fiber-rich foods. The inadequate intake of fiber is concerning because dietary fiber is vital for maintaining digestive health, preventing constipation, and managing chronic diseases like diabetes and cardiovascular conditions [49,50,51,52,53]. Addressing all these nutritional gaps is crucial for promoting healthy aging and preventing age-related health complications.

## 3. Screening Possible Solutions

To address nutritional gaps in the senior population, a comprehensive approach should target bridging gaps in macronutrients, micronutrients, and non-nutrients. For macronutrients, protein enrichment has been deemed crucial, with recommendations of 1.0–1.2 g protein per kilogram of body weight daily [7,54,55,56]. The guidelines emphasize easily digestible forms of food proteins, leucine-rich sources, caloric-dense foods and incorporating healthy fats like omega-3s; all with the aim of supporting muscle maintenance, tackling sarcopenia of aging and meeting elderly energy needs [2]. Furthermore, increasing dietary fiber intake is crucial for digestive health, bowel regularity, and managing chronic conditions common in seniors [49,57].

Micronutrient fortification should focus on identified deficiencies of vitamins and minerals, with current particular attention to vitamin D (800 IU daily) and calcium (1200 mg/day) for those over 70 years [37,58]. Moreover, food delivery systems and encapsulation platforms could help enhance absorption, stability and compatibility to various food applications [59,60]. For non-nutrients, phytochemical-rich ingredients and food antioxidants like carotenoids and flavonoids can boost antioxidant intake with potential benefits to consumer health and alleviation of age-related symptoms [61,62,63]. Specific compounds such as macamides from maca root may offer benefits such as enhanced fertility, alleviation of menopausal symptoms, and improved mood and energy levels [64,65,66]. Furthermore, olive leaf extract, containing oleuropein as its primary active ingredient, reduces mitochondrial superoxide production and increases mitochondrial biogenesis, contributing to overall cellular health [67,68]. Thus, developing functional foods or supplements that combine beneficial nutrients and non-nutrients may offer a comprehensive solution to support seniors’ health. Yet, the success of such a solution requires that, in addition to meeting consumer needs, it is formulated into a palatable product corresponding to consumer liking and preferences and acceptance.

## 4. Principles for Designing Novel Food Solutions: The EAT4AGE Menu of Choices

The development of nutritional solutions for seniors presents a complex challenge, beginning with a fundamental strategic decision: should we pursue the development of a groundbreaking innovative product, or focus on reformulating existing products that are accepted by seniors? This choice between creating novel offerings and refining familiar ones is not straightforward, as it involves considering a multitude of factors. These include market needs, consumer culture and behavior, technological capabilities, specific nutritional requirements of seniors, sensory preferences, ease of use, and potential health impacts. Each of these elements plays a crucial role in determining the commercial success, clinical effectiveness and consumer adoption of the final product.

To navigate this complexity and ensure a comprehensive approach to product development, we have outlined a detailed checklist of considerations that should be taken into account during prototyping an age-tailored food (see Figure 3). This tool serves as a structured guide, outlining the critical factors and considerations that must be addressed throughout the development process. It advocates for a four-step paradigm for prototype development from ideation to field tests and commercialization. These are centered around six main pragmatic pillars, from considering product and process attributes up to regulatory and societal factors. Each pillar encompasses various aspects that should be carefully addressed during product development. For example, the product characteristics pillar invites professionals to visit considerations such as nutritional adequacy, safety, sensory optimization, packaging design, portion size and ease of use—all tailored to the unique needs of the consumer, i.e., older adults.

By systematically addressing each point on the pillars of the checklist, developers can ensure that they are creating products that not only meet the nutritional needs of seniors but also align with their preferences and capabilities as well as the regulatory and societal settings. This methodical approach helps bridge the gap between scientific research on elderly nutrition and the practical application of this knowledge in product development, ultimately leading to more effective and well-received nutritional solutions for the senior population. Moreover, it ensures developers do not overlook other various pragmatic considerations, such as regulatory or sustainability considerations.

Furthermore, it is crucial to recognize that meeting the needs of older adults extends beyond food safety, nutritional adequacy, taste requirements and digestive capacity. In fact, adoption of novel and/or functional foods should also address the food preferences of elderly people, which are significantly influenced by factors such as personal habits, regional differences, and cultural backgrounds [69,70,71]. Therefore, nutritional strategies to design foods for seniors should seek to develop a diverse menu of choices that could address various consumers. The EAT4AGE project adapted the paradigm outlined in Figure 3 to effectively cater to the development of four healthier food choices designed for seniors: a cereal-based product, reformulated dairy products, a processed meat product and an enriched spread.

In general, development of an innovative product should include meticulous sensory optimization to ensure palatability alongside careful analysis of consumer liking and willingness to try the product. Additionally, the practicality of portion size and ease of use (e.g., ease of opening packaged food) should be accounted for. Conversely, when imitating or reformulating an existing product, adaptations must be made to suit the unique needs of seniors while preserving the product’s typical characteristics. This includes adjusting textures to facilitate easier chewing and swallowing, modifying flavors to maintain appeal despite age-related taste changes, and nutritional fortification to meet senior-specific dietary requirements. As part of this process, the EAT4AGE consortium initiated an extensive literature review that led to defining the age threshold for older adults as 65 years and above, aligning with gerontological categorizations and acknowledging that physiological age, rather than chronological age, is more relevant for digestive processes in this population [9,72,73,74]. Recognizing the paramount importance of food palatability, product acceptability was evaluated through a dual approach: untrained consumer panels in the country of development and expert sensory analysis at Nofima. This method accounts for sociocultural influences on food preferences, particularly among older adults [17,69,71,75]. Moreover, this approach aligns with the UN’s Sustainable Development Goal 3, emphasizing culturally appropriate nutritional interventions [12].

### 4.1. Co-Extruded Plant-Based Cereal

One of the primary outcomes of the EAT4AGE project was a co-extruded plant-based functional cereal designed to bridge the gap between conventional cereals and the nutritional requirements of older adults. This product was premeditated to offer high caloric value, high protein content to support muscle mass maintenance, and elevated fiber content. We emphasized the development of plant-based products to provide a balanced nutrient profile, aligning with global health recommendations to increase consumption of plant-based foods [3,12,76]. This product was engineered to have high protein content using a blend of plant-based proteins, boosted fiber levels, minimal added sugars, and incorporating bioactive antioxidants like Maca powder and Olive Leaf Extract [64,65,66,67,68]. Prototype production at pilot scale was achieved using co-extrusion technology, leading to fabrication of a softer cereal structure, and incorporating a fatty sesame paste filling. A cohort of untrained elderly consumers (n = 21, mean age 73 ± 5 years) evaluated the product, demonstrating high palatability across multiple sensory parameters and overall acceptability. These findings were subsequently validated by a trained sensory panel, providing corroborative evidence for the product’s sensory qualities. This product was found to be of high caloric density, surpassing commercially available products and exhibiting high in vitro digestibility and good sensorial properties [77].

### 4.2. Reformulated Dairy Products

Diet-induced muscle mass and strength loss in older adults may be due to insufficient protein intake [32]. Therefore, older adults need to increase the amount of high-quality ingested proteins, particularly foods rich in leucine, to promote muscle health [54]. A strategy to stimulate muscle protein synthesis in older adults is to increase the intake of branched amino acids, like leucine, through dairy product enrichment in whey proteins. However, adding whey proteins may modify the properties of traditional foods such as yogurt or cheese (e.g., firmer texture, increased astringency), which may in turn influence food intake and nutrient digestibility. In addition, older adults need to maintain a high calcium intake to avoid osteoporosis, and dairy foods are an excellent source of calcium. Age-tailored foods can be further fortified with calcium, but not all sources of calcium have the same bioaccessibility and it must be selected carefully [78]. Within the framework of the EAT4AGE project, two age-tailored dairy products were designed and manufactured: a fermented dairy dessert and a spreadable cream cheese. These products were enriched in total protein compared to their traditional versions (+150% for the yogurt-type dessert, and +300% for the cream cheese), and were formulated with a ratio of whey proteins to caseins of 80 to 20% (as opposed to milk). This combination of whey proteins and caseins was chosen with the intention of promoting a prolonged postprandial hyperaminoacidemia to efficiently stimulate muscle synthesis. The cream cheese product was also supplemented in calcium carbonate (0.5 *w*/*w*). Lipid content and heat treatment time/temperature were optimized to obtain products with desirable physical and sensory characteristics to comply with older adults’ eating habits and abilities.

The rheological properties of the products obtained were like those of traditional products marketed in France [79], and according to a panel of 80 subjects aged 76 ± 6 y. (49 women, 31 men), both products were “easy to eat” and obtained high ratings for oral comfort (i.e., > 75/100), among other food comfortability attributes (data not published). However, results from in vitro digestion experiments suggest that proteolysis may be less efficient in whey protein-based products than in casein-based products [79,80], particularly when enzyme activities are decreased, as observed in the gastrointestinal tract of older adults [9]. A clinical study is underway to confirm these observations in vivo but, in any case, these results highlight the fact that the food microstructure and digestibility must be carefully considered to develop whey protein-rich dairy products adapted to the nutritional needs of older adults.

### 4.3. Tenderized Processed Meat

Meat, a significant source of high-quality protein and essential amino acids, has been a fundamental component of human nutrition throughout evolutionary history [81,82]. Its nutritional profile renders it particularly suitable for both geriatric and younger populations [82]. However, the consumption of certain meat cuts can present challenges for older individuals due to the presence of tough connective tissue surrounding muscle fibers, which may impede mastication and deglutition processes [83].

To address this issue, mechanical tenderization was explored as a means to improve meat palatability for elderly consumers. The research utilized *M. semitendinosus* muscle samples obtained from Norwegian Red cattle (*Bos taurus*). The experimental design involved a comparative analysis between mechanically tenderized meat supplemented with a protein hydrolysate and untreated controls. In the methodology, muscle samples were divided into two cohorts: a treatment group subjected to blade tenderization and brine injection with protein hydrolysate, and a control group left untreated. Both cohorts underwent identical thermal processing to ensure consistency. A sensory evaluation was conducted using a panel of 50 elderly consumers (n = 50) in a double-blind taste test, where samples were presented without information regarding preparation methods.

### 4.4. Protein-Enriched Desserts

Dessert is consumed by several older adults as a habit [84]. Older people place a high value on desserts because of their palatability and sensory qualities [85]. For their nutritional needs, older people are less inclined to use high-protein shakes or novel products; instead, they prefer to eat foods they are already familiar with [84]. Dairy treats are therefore well liked by senior citizens. Desserts made with milk add to the overall caloric density of a meal, in addition to being comfort foods. Starting from an existing product, dulce de leche, consisting of 6–7% protein as a standard, our goal was to create a high-protein milk-based dessert. The developed high-protein product had a protein content of 20% (*w*/*w*) and 4 times the leucine content compared to the control. The high leucine content and high digestion of whey protein isolate made it a useful ingredient for protein enrichment. Additionally, the high protein dessert has high culinary versatility and can be used as a spread on bread, crepes or as a dessert accompaniment. Alternatively, the product is pleasant to be consumed alone, and a serving of 25 g would provide about 6 g of protein. When taken together with a main meal, the protein from the dessert would substantially augment the protein load of the meal, which is typically expected to be between 25–30 g [86].

The EAT4AGE project provides valuable insights into developing nutritional solutions for older adults, but several limitations warrant consideration and/or future work. The study’s focus on European populations may limit its applicability to regions with diverse dietary habits and cultural contexts, as food choices of elderly individuals are significantly influenced by personal habits, regional variations, and cultural backgrounds [17,69,70,71]. While the study validated the potential for scaling nutritional solutions for seniors, it was conducted on a small scale and did not reach the scale-up phase. The economic feasibility and marketability of these specialized food products were not extensively explored, potentially affecting their accessibility to the target population. In fact, future work should carefully consider the question of how food for seniors could be effectively marketed to this stratum of the population, which would not like to view itself as elderly. Another notable limitation is the limited number of clinical trials on functional foods for seniors, which restricts the ability to draw definitive conclusions about their long-term efficacy in improving health outcomes for older adults. Moreover, little is known about the possible food–drug interplay in older adults, particularly as it pertains to functional and/or novel foods. These limitations underscore areas for much-needed future research in face of the growing aging population.

## 5. Conclusions

The global challenge to align life expectancy with health span emphasizes a critical need for targeted nutrition strategies to support healthy aging. As aging populations continue to grow worldwide, malnutrition and hidden hunger among seniors emerge as pressing concerns, contributing to increased risks of chronic diseases and impacting overall well-being. Innovative food solutions that cater specifically to the nutritional deficiencies prevalent among older adults—such as insufficient protein, fiber, and antioxidants—can play a transformative role in enhancing their overall health and well-being. Thus, a systematic paradigm for food product development could offer an avenue to creating accessible, appealing, and nutrient-rich food products. This science- and evidence-driven approach will help empower seniors to make healthier dietary choices. Moreover, it will feed food professionals with a systematic scheme to develop consumer-oriented food choices, thereby improving quality of life and reducing healthcare and societal burdens associated with aging.

## Figures and Tables

**Figure 1 foods-14-00396-f001:**
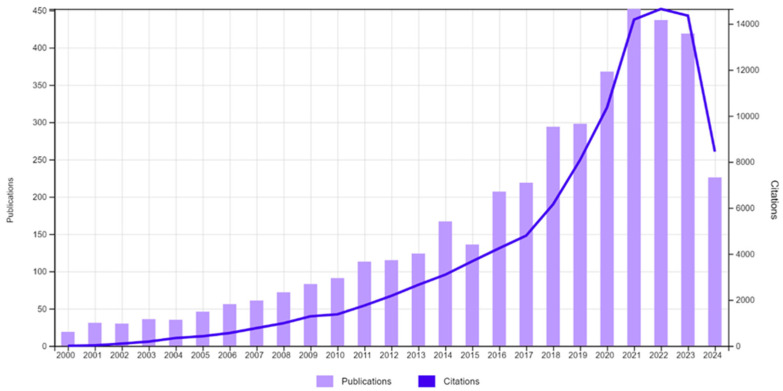
Literature survey of scientific publications and citations between 2000–2024 related to the general theme of foods for the elderly and healthy aging. (Source: Clarivate Web of Science, Accessed on 24 July 2024).

**Figure 2 foods-14-00396-f002:**
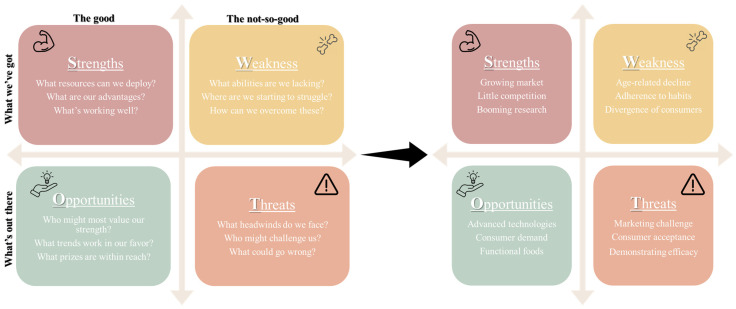
SWOT analysis of developing foods tailored for older adults from a general approach to a more specific example of the EAT4AGE analysis.

**Figure 3 foods-14-00396-f003:**
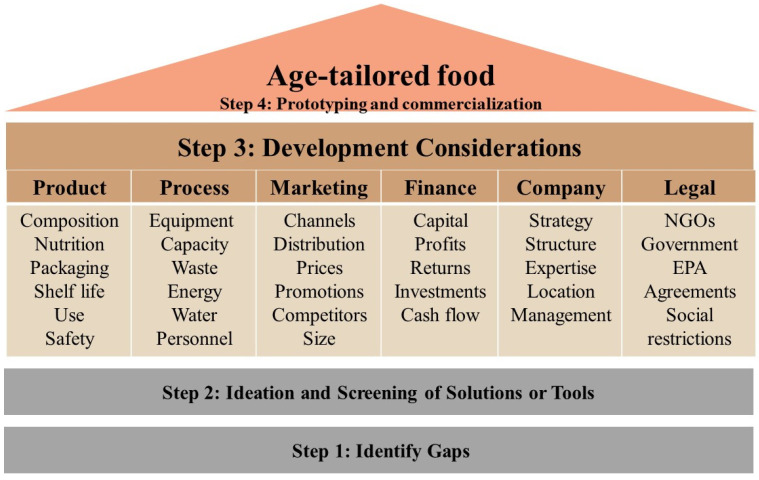
Considerations for food product development: foundation, columns, and roof for success. The foundation involves understanding gaps and potential solutions. The columns represent the various critical areas to consider, while the roof symbolizes the final product.

## Data Availability

Data is available upon request from the corresponding author.

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
