# Peer review of "Strategic Considerations in Designing Food Solutions for Seniors"

_foods, 2025, doi:10.3390/foods14030396_

Round 1

Reviewer 1 Report

Comments and Suggestions for Authors

It is very important to develop a food processing plan suitable for the elderly. This paper provides a series of strategic considerations for designing senior food,which are feasible and meaningful.  Besides meeting the nutritional needs, taste requirements, digestive capacity, and ensuring food safety, the preferences of elderly people for food may be influenced by factors such as personal habits, regional differences, and cultural backgrounds.

Therefore, in order to meet the needs of the elderly, food processing plans should not only provide various flavors and taste choices, but also consider the dietary preferences and cultural background of the elderly. By fully considering the special needs and backgrounds of the elderly, developing food processing plans suitable for them can provide strong support for their health and quality of life. I suggest the authors can supplement these contents in this manuscript.

Author Response

Please see attached a details point by point response to reviewers' reports

Reviewer 2 Report

Comments and Suggestions for Authors

Dear Authors, 

The goal of the submitted manuscript entitled “Strategic Considerations in Designing Food Solutions for Seniors” was to offer insight into the current nutritional needs of seniors and introduce a tool developed to facilitate new product development aimed at this population group.

In order to increase the article’s clarity I suggest to introduce the following changes: 

1. add more details about the EAT4AGE project, explaining  who was considered a “senior” and where were the new products/solutions tested. This is important because the acceptability of and desirability for foods is heavily influenced by sociocultural factors – cite more on this, also referring to SDG3 (for example) in https://doi.org/10.1177/0379572120975874).

Regarding the SWOT analysis – consider creating a separate chapter about it, including a paragraph on SWOT methodology. Several  elements presented in Fig. 2 are missing therefore need to be explained – for example  “little competition” (one of the identified strengths). 

2. Chapter 2, although short, is probably the most comprehensive part of the manuscript. What is however quite conspicuous is the number of times Reference 2 (Redmond et al., 2015 is cited 9x in Chapter 2 alone, overall 16x in the manuscript!) The 2015 publication in Oncotarget has 454 references – shouldn't some of them (as primary sources) be rather cited, rather than  Remond et al.?

4. merge chapters 4 (Principles for Designing Novel Food Solutions) and 5 (The Eat4AGE Menu of Choices) to show how the tool was (or can be) used to develop age-tailored foods. This would increase the soundness of the paper.  The abstract implies that the Authors will: (…) describe the application of the tool in a the development of a series of products (…). 

L237 (chapter 5.1) – hasn't the paper in Food Science and Human Wellness been published yet?

In an attempt to improve the article I would also like to point out that the term “nutritional interventions” (L12, L40, L208) is rather linked to medical, not public health (preventing malnutrition) and marketing contexts. Also: please describe any limitations of your research.

Author Response

Please see attached a detailed point by point response to reviewers' reports

Round 2

Reviewer 2 Report

Comments and Suggestions for Authors

Dear Authors, your changes have - in my opinion - improved the clarity of the  article and have made reading through the description of the conducted SWOT analysis easier. I would like to repeat my suggestion to delete Fig. 2, SWOT external and internal factors determinants (L101). 

As you aim to improve the manuscript please add all names of authors in the list of references.

Author Response

Please see attached responses
